# BAYESIAN LEARNING TO OPTIMIZE:
# QUANTIFYING THE OPTIMIZER UNCERTAINTY

## ABSTRACT

Optimizing an objective function with uncertainty awareness is well-known to improve the accuracy and confidence of optimization solutions. Meanwhile, another relevant but very different question remains yet open: how to model and quantify the uncertainty of an optimization algorithm itself? To close such a gap, the prerequisite is to consider the optimizers as sampled from a distribution, rather than a few pre-defined and fixed update rules. We first take the novel angle to consider the algorithmic space of optimizers, each being parameterized by a neural network. We then propose a Boltzmann-shaped posterior over this optimizer space, and approximate the posterior locally as Gaussian distributions through variational inference. Our novel model, *Bayesian learning to optimize* (**BL2O**) is the first study to recognize and quantify the uncertainty of the optimization algorithm. Our experiments on optimizing test functions, energy functions in protein-protein interactions and loss functions in image classification and data privacy attack demonstrate that, compared to state-of-the-art methods, BL2O improves optimization and uncertainty quantification (UQ) in aforementioned problems as well as calibration and out-of-domain detection in image classification.

## 1 INTRODUCTION

Computational models of many real-world applications involve optimizing non-convex objective functions. As the non-convex optimization problem is NP-hard, no optimization algorithm (or optimizer) could guarantee the global optima in general, and instead, their solutions' usefulness (sometimes based on their proximity to the optima), when the optima are unknown, can be very uncertain. Being able to quantify such uncertainty is important to not only assessing the solution uncertainty after optimization but also enhancing the search efficiency during optimization. For instance, reliable and trustworthy machine learning models demand uncertainty awareness and quantification during training (optimizing) such models, whereas in reality deep neural networks without proper modeling of uncertainty suffer from overconfidence and miscalibration (Guo et al., 2017). In another application example of protein docking, although there exists epistemic uncertainty of the objective function and the aleatoric uncertainty of the protein structure data (Cao & Shen, 2020), state-of-the-art methods only predict several single solutions (Porter et al., 2019) without any associated uncertainty, which makes those predictions hard to be interpreted by the end users.

Various optimization methods have been proposed in response to the need of uncertainty awareness. Stochastic optimization methods like random search (Zhigljavsky, 2012), simulated annealing (Kirkpatrick et al., 1983), genetic algorithms (Goldenberg, 1989) and particle swarm optimization (Kennedy & Eberhart, 1995) injected the randomness into the algorithms in order to reduce uncertainties. However, these methods do not provide the uncertainty quantification (UQ) of solutions. Recently, there have been growing interests in applying inference-based methods to optimization problems (Brochu et al., 2010; Shapiro, 2000; Pelikan et al., 1999). Generally, they transfer the uncertainties within the data and model into the final solution by modelling the posterior distribution over the global optima. For instance, Bijl et al. (2016) uses sequential Monte Carlo to approximate the distribution over the optima with Thompson sampling as the search strategy. Hernández-Lobato et al. (2014) uses kernel approximation for modelling the posterior over the optimum under Gaussian process. Ortega et al. (2012); Cao & Shen (2020) directly model the posterior over the optimum as a Boltzmann distribution. They not only surpass the previous methods in accuracy and efficiency, but also provide easy-to-interpret uncertainty quantification.

Despite progress in optimization with uncertainty-awareness, significant open questions remain. Existing methods consider uncertainty either within the data or the model (including objective functions) (Kendall & Gal, 2017; Ortega et al., 2012; Cao & Shen, 2020). *However, no attention was ever paid to the uncertainty arising from the optimizer that is directly responsible for deriving the end solutions with given data and models.* The optimizer is usually pre-defined and fixed in the optimization algorithm space. For instance, there are several popular update rules in Bayesian optimization, such as expected improvement Vazquez & Bect (2010) or upper confidence bound Srinivas et al. (2009), that are chosen and fixed for the entire process. For Bayesian neural networks training, the update rule is usually chosen off-the-shelf, such as Adam, SGD, or RMSDrop. *The uncertainty in the optimizer is intrinsically defined over the optimizer space and important to the optimization and UQ solutions. However, such uncertainty is unwittingly ignored when the optimizer is treated as a fixed sample in the space.*

To fill the aforementioned gap, the **core intellectual value** of this work is to recognize and quantify a new form of uncertainty, that lies in the optimization algorithm (optimizer), besides the classical data- or model- based uncertainties (also known as epistemic and aleatoric uncertainties). The **underlying innovation** is to treat an optimizer as a random sample from the *algorithmic space*, rather than one of a few hand-crafted update rules. The **key enabling technique** is to consider the *algorithmic space* being parameterized by a neural network. We then leverage a Boltzmann-shaped posterior over the optimizers, and approximate the posterior locally as Gaussian distributions through variational inference. Our approach, *Bayesian learning to optimize* (**BL2O**), **for the first time** addresses the modeling of the optimizer-based uncertainty. Extensive experiments on optimizing test functions, energy functions in a bioinformatics application, and loss functions in the image classification and data privacy attack demonstrate that compared to the start-of-art methods, BL2O substantially improves the performance of optimization and uncertainty quantification, as well as calibration and out-of-domain detection in classification.

In the following sections, we first review related methods in details and reveal the remaining gap. We then formally define the problem of optimization with uncertainty quantification and point out the optimizer as a source of uncertainty. After formally defining the optimizer space, the optimal optimizer as a random vector in the space, and the optimizer uncertainty, we propose our novel model, **BL2O**. And lastly, we compare our **BL2O** with both Bayesian and non-Bayesian competing methods on extensive test functions and real-world applications.

## 2 RELATED WORK

Many works (Wang & Jegelka, 2017; Hennig & Schuler, 2012) studied optimization with uncertainty quantification under the framework of Bayesian optimization (Shahriari et al., 2016; Brochu et al., 2010). In these studies, multiple objectives are sampled from the posterior over the objectives ($p(f|\mathcal{D})$), where $\mathcal{D}$ is the observed data. Each sampled objective is optimized for obtaining samples of the global optima: $\boldsymbol{w}^*$ so that the empirical distribution over $\boldsymbol{w}^*$ can be built. Approximation is much needed since those approaches need optimization for every sample. For instance, Henrández-Lobato et al. (2014) uses kernel approximation to approximate the posterior distirbution.

Another line of work uses various sampling schemes for estimating the density of posterior distributions. For instance, Bijl et al. (2016) uses sequential Monte Carlo sampling. De Bonet et al. (1997) designs a randomized optimization algorithm that directly samples global optima. These methods are much more efficient, but their performance heavily depends on the objective landscapes. Moreover, a few studies (Ahmed et al., 2016; Lizotte, 2008; Osborne et al., 2009; Wu et al., 2017) in Bayesian optimization utilize first-order information to boost the performance of optimization. For instance, Osborne et al. (2009) uses gradient information to improve the covariance matrix in Gaussian process. Wu et al. (2017) embeds the derivative knowledge into the acquisition function which is optimized in every iteration.

Finally, there are approaches (Ortega et al., 2012; Cao & Shen, 2020) that directly model the shape of posterior as the Boltzmann distributions: $p(\boldsymbol{w}^*|D) \propto \exp(-\alpha f(\boldsymbol{w}^*))$, where $\alpha$ is the scheduled temperature constant. They automatically adjust $\alpha$ during the search in order to balance the exploration-exploitation tradeoff. They beat previous work in terms of both efficiency and accuracy.

However, as revealed earlier in the Introduction, none of the methods above consider the uncertainty within the optimizer.

## 3 METHODS

**Notation**. We use a bold-faced uppercase letter to denote a matrix (e.g. $\boldsymbol{W}$), a bold-faced lowercase letter to denote a vector (e.g. $\boldsymbol{w}$), and a normal lowercase letter to denote a scalar (e.g. $w$).

### 3.1 PROBLEM STATEMENT

The goal of optimization is to find the global optimum for an objective function $f(\boldsymbol{w})$ w.r.t. $\boldsymbol{w}$:

$$\boldsymbol{w}^* = \arg\min_{\boldsymbol{w}} f(\boldsymbol{w}). \tag{1}$$

$\boldsymbol{w}^*$ is assumed unknown and treated as a random vector in this study. Once a optimizer obtains $\hat{\boldsymbol{w}}$, its estimate of $\boldsymbol{w}^*$, it is important to assess the quality and the uncertainty of the solution. Considering that many real-world objective functions are nonconvex and noisy in $\hat{\boldsymbol{w}}$, solution quality is often measured by $||\hat{\boldsymbol{w}} - \boldsymbol{w}^*||$, the proximity to the global optimum rather than that to the optimal function value. Examples include energy functions as the objective and RMSDs as the proximity measure in protein docking (Lensink et al., 2007). Therefore, the goal of uncertainty quantification (UQ) here is the following:

$$P(||\hat{\boldsymbol{w}} - \boldsymbol{w}^*|| \leqslant r_\sigma | \mathcal{D}) = \sigma \tag{2}$$

where $r_\sigma$ is the upper bound of $||\hat{\boldsymbol{w}} - \boldsymbol{w}^*||$ at $\sigma$ confidence level, and $\mathcal{D}$ denotes samples during optimization. Such UQ results additionally provide confidence in the solution $\hat{\boldsymbol{w}}$ and improve model reliability for end users.

To calculate the probability defined in Eq 2 and perform UQ, a direct albeit challenging way is to model the posterior over $\boldsymbol{w}^*$ ($p(\boldsymbol{w}^*|\mathcal{D})$) and then sample from the posterior. When the optimizer $g$ is regarded fixed in existing literature, the posterior is actually $p(\boldsymbol{w}^*|\mathcal{D}, g)$. A central contribution of ours is to further consider the optimizer as a source of uncertainty, model it as a random vector in an optimizer space, and perform posterior estimation of $p(\boldsymbol{w}^*|\mathcal{D})$.

### 3.2 OPTIMIZER UNCERTAINTY: A FRAMEWORK

An optimizer is directly responsible for optimization and thus naturally a source of solution uncertainty. To address this often-neglected uncertainty source, we first define the space of optimizers and then model an optimizer as a point in this space. Considering that many widely-used optimizers are iterative and using first-order derivatives, we restrict the optimizer space as follows:

**Definition 3.1 ((First-order Iterative) Optimizer Space)** *We define a first-order, iterative algorithmic space $\mathcal{G}$, where each point $g \in \mathcal{G}$ is an iterative optimizer, that has the following mapping: $g(\{\nabla f(\boldsymbol{w}^\tau)\}_{\tau=1}^t) = \delta\boldsymbol{w}^t$, where $\nabla f(\boldsymbol{w})^\tau$ and $\delta\boldsymbol{w}^t$ are the gradient and the update vector at $\tau$th and $t$th iteration, respectively.*

Here we use $g(\cdot)$ to denote a pre-defined update rule and the resulting optimizer. For instance, in gradient descent, $g(\{\nabla f(\boldsymbol{w}^\tau)\}_{\tau=1}^t) = -\alpha\nabla f(\boldsymbol{w}^t)$, where $\alpha$ is the step size. Now that the optimizer space is defined, we next define the (unknown) optimal optimizer and its uncertainty.

**Definition 3.2 (Optimal Optimizer)** *We define the optimal optimizer $g^* \in \mathcal{G}$ as the optimizer that can obtain the lowest function value with a fixed budget $T$:*

$$g* = \arg\min_{g \in \mathcal{G}} (\sum_{t=1}^T f(\boldsymbol{w}_g^t)) \tag{3}$$

*where $\boldsymbol{w}_g^t = \boldsymbol{w}_g^{t-1} + g(\{\nabla f(\boldsymbol{w}_g^\tau)\}_{\tau=1}^{t-1})$ is the parameter value at $t$th iteration updated through the optimizer g.*

In practice the optimal optimizer $g^*$ is unknown so we treat $g^*$ as a **random vector** and formally define the optimizer uncertainty as follows:

**Definition 3.3 (Optimizer Uncertainty)** *Let $\mathcal{G}$ be the algorithmic space, where each point $g \in \mathcal{G}$ is an optimizer. We assume there is a prior distribution over the optimal optimizer $g^*$ as $p(g^*)$. We also assume a likelihood distribution as $p(\mathcal{D}|g*)$, where $\mathcal{D}$ are the observed data (sample trajectory) given $g^*$. Then we define the optimizer uncertainty through $p(g^*|\mathcal{D}) \propto p(\mathcal{D}|g^*)p(g^*)$.*

To inject the optimizer uncertainty into $p(\boldsymbol{w}^*|\mathcal{D})$, it is straightforward to have the following integration for posterior estimation:

$$p(\boldsymbol{w}^*|\mathcal{D}) = \int p(g^*|\mathcal{D})p(\boldsymbol{w}^*|\mathcal{D}, g^*)dg \tag{4}$$

### 3.3 Parameterizing the Optimizer Space

The optimizer uncertainty $p(g^*|\mathcal{D})$ as defined in Def. 3.3 can be intractable when there is no proper parameterization of the optimizer space $\mathcal{G}$. Therefore, we next introduce possible ways of parameterizing $\mathcal{G}$ as defined in Def. 3.1.

**Parameterization through Hyperparameters of Specific Optimizers.** A simple way to parameterize the optimizer space for classical optimizers (e.g. Gradient Descent, Adam) is based on their hyperparameters: $\mathcal{G} = \mathcal{H}$, where $\mathcal{H}$ is the hyperparameter space. For instance, for gradient descent, we have $\mathcal{H} = (\alpha)$, where $\alpha$ is the learning rate. For Adam, we have $\mathcal{H} = (\alpha, \beta_1, \beta_2)$, where $\beta_1$ and $\beta_2$ are the coefficients used for computing running averages of gradient and its square.

However, such parameterization has significant drawbacks. The resulting algorithmic space $\mathcal{G}$ is very limited and heavily depends on the specific optimizer. The $\mathcal{G}$ (a 1D space) parameterized by the hyperparameters of gradient descent is different from that (a 3D space) parameterized by the hyperparameters of Adam. In fact, each is a rather restricted region of the actual $\mathcal{G}$. The intrinsic flexibility (uncertainty) that lies in an iterative optimizer's update rule is not explored at all in this parameterization. These drawbacks are empirically demonstrated in Sec. 4.

**Parameterization through Neural Networks.** In order to reasonably and accurately model the intrinsic uncertainty within the update rule, we need to find a much more flexible way for modelling $g$. We thus consider to parameterize the optimizer space as a neural network: $\mathcal{G} = \Theta$, where each $\boldsymbol{\theta} \in \Theta$ are the parameters in the neural network. Overcoming drawbacks of the optimizer space $\mathcal{H}$ by hyperparameters, $\Theta$ by neural network parameters generalizes update rules through neural networks that can represent a wide variety of functions. We note that this is also the space of meta-optimizers that learn to optimize (L2O) iterative update rules from data on a given task (Andrychowicz et al., 2016; Chen et al., 2017; Lv et al., 2017; Cao et al., 2019a). However, there has been no notion of uncertainty let alone the task of UQ for the learned optimizer in these L2O methods, which is to be addressed in our Bayesian L2O (BL2O).

### 3.4 Modeling an optimizer as a random vector

Now that we have the optimizer space $\mathcal{G}$ properly defined and parameterized, we proceed to model an optimizer $g$ as a random vector in the space.

**Boltzmann-shaped Posterior.** Since we have $\mathcal{G} = \Theta$, we can rewrite each $g \in \mathcal{G}$ as $g_{\boldsymbol{\theta}}$ with $\boldsymbol{\theta} \in \Theta$ and the optimal optimizer $g^*$ as $g_{\boldsymbol{\theta}^*}$. Therefore, $p(g^*|\mathcal{D})$ becomes $p(\boldsymbol{\theta}^*|\mathcal{D})$. We consider a Gaussian prior over the parameters of the neural network : $p(\boldsymbol{\theta}^*) \propto \exp(-\lambda||\boldsymbol{\theta}^*||_2^2)$, where $\lambda$ is a constant controlling the variance. We use the chain rule to decompose the likelihood function $p(\mathcal{D}|\boldsymbol{\theta}^*)$ at a fixed budge $T$:

$$p(\mathcal{D}|\boldsymbol{\theta}^*) = \prod_{t=1}^{T} p(f(\boldsymbol{w}_{\boldsymbol{\theta}^*}^t), \boldsymbol{w}_{\boldsymbol{\theta}^*}^t|\boldsymbol{\theta}^*, \{f(\boldsymbol{w}_{\boldsymbol{\theta}^*}^\tau), \boldsymbol{w}_{\boldsymbol{\theta}^*}^\tau\}_{\tau=0}^{t-1}) = \prod_{t=1}^{T} p(f(\boldsymbol{w}_{\boldsymbol{\theta}^*}^t), |\boldsymbol{w}_{\boldsymbol{\theta}^*}^t, \boldsymbol{\theta}^*, \{f(\boldsymbol{w}_{\boldsymbol{\theta}^*}^\tau), \boldsymbol{w}_{\boldsymbol{\theta}^*}^\tau\}_{\tau=0}^{t-1})$$

$$\tag{5}$$

The second equality is due to that $\boldsymbol{w}_{\boldsymbol{\theta}^*}^t$ is fixed given $\boldsymbol{\theta}^*$ and past data points. For the single sample likelihood, we apply the results from Ortega et al. (2012); Cao & Shen (2020) and obtain

$$p(f(\boldsymbol{w}_{\boldsymbol{\theta}^*}^t), |\boldsymbol{w}_{\boldsymbol{\theta}^*}^t, \boldsymbol{\theta}^*, \{f(\boldsymbol{w}_{\boldsymbol{\theta}^*}^\tau), \boldsymbol{w}_{\boldsymbol{\theta}^*}^\tau\}_{\tau=1}^{t-1}) \propto \exp(-f(\boldsymbol{w}_{\boldsymbol{\theta}^*}^t)) \tag{6}$$

We multiply the likelihood functions of all samples together and obtain the Boltzmann-shaped likelihood function as $p(\mathcal{D}|\boldsymbol{\theta}^*) \propto \exp(-\sum_{t=1}^{T} f(\boldsymbol{w}_{\boldsymbol{\theta}^*}^t))$. We finally multiply the conjugate prior to the likelihood and obtain the Boltzmann-shaped posterior as:

$$p(\boldsymbol{\theta}^*|\mathcal{D}) \propto \exp(-\sum_{t=1}^{T} f(\boldsymbol{w}_{\boldsymbol{\theta}^*}^t)) \cdot \exp(-\lambda||\boldsymbol{\theta}^*||_2^2) = \exp(-F(\boldsymbol{\theta}^*)) \tag{7}$$

where $F(\boldsymbol{\theta}^*) = \sum_{t=1}^{T} f(\boldsymbol{w}_{\boldsymbol{\theta}^*}^t) + \lambda||\boldsymbol{\theta}^*||_2^2$, which actually contains the objective in Eq 3 plus a L2 regularization constant.

**Local Approximation and Bayesian Loss.** However, the above posterior distribution involves an integral in the normalization constant which is computationally intractable. Moreover, the architecture of $F(\boldsymbol{\theta}^*)$ is so complicated that it is impossible to directly sample from the posterior distribution. In order to overcome the aforementioned challenges, we would like to learn a distribution function $q(\boldsymbol{\theta}^*|\boldsymbol{\phi})$ that has the analytic form and is easy to be sampled, where $\boldsymbol{\phi}$ is the parameter vector in $q(\boldsymbol{\theta}^*|\boldsymbol{\phi})$, to approximate the real posterior $p(\boldsymbol{\theta}^*|\mathcal{D})$.

Furthermore, due to the high dimensions of $\boldsymbol{\theta}^*$ and the complicated landscape of the posterior, it is impossible to approximate $p(\boldsymbol{\theta}^*|\mathcal{D})$ at every position in the $\boldsymbol{\theta}^*$ space. We then consider to approximate it locally around $\boldsymbol{\theta}^c$, an optimum of interest for $F(\boldsymbol{\theta}^*)$.

We denote the local region as $\Theta^c$, a neighborhood around $\boldsymbol{\theta}^c$, and re-normalization constant $C = \int_{\boldsymbol{\theta}^* \in \Theta^c} p(\boldsymbol{\theta}^*|\mathcal{D})d\boldsymbol{\theta}^*$. Then the local posterior will be a conditioned (re-scaled) version of $p(\boldsymbol{\theta}^*|\mathcal{D})$: $p'(\boldsymbol{\theta}^*|\mathcal{D}) = p(\boldsymbol{\theta}^*|\mathcal{D})/C$, $\boldsymbol{\theta}^* \in \Theta^c$. In order to make $q(\boldsymbol{\theta}^*|\boldsymbol{\phi}) \approx p'(\boldsymbol{\theta}^*|\mathcal{D})$, we calculate the KL-divergence between these two:

$$
\begin{aligned}
\mathrm{KL}(q(\boldsymbol{\theta}^*|\boldsymbol{\phi})||p'(\boldsymbol{\theta}^*|\mathcal{D})) &= \int_{\boldsymbol{\theta}^* \in \Theta^c} q(\boldsymbol{\theta}^*|\boldsymbol{\phi}) \log \frac{q(\boldsymbol{\theta}^*|\boldsymbol{\phi})}{p'(\boldsymbol{\theta}^*|\mathcal{D})} d\boldsymbol{\theta}^* = \int_{\boldsymbol{\theta}^* \in \Theta^c} q(\boldsymbol{\theta}^*|\boldsymbol{\phi}) \log \frac{q(\boldsymbol{\theta}^*|\boldsymbol{\phi})}{p(\boldsymbol{\theta}^*|\mathcal{D})/C} d\boldsymbol{\theta}^* \\
&= \int_{\boldsymbol{\theta}^* \in \Theta^c} q(\boldsymbol{\theta}^*|\boldsymbol{\phi}) \log \frac{q(\boldsymbol{\theta}^*|\boldsymbol{\phi})}{\exp(-F(\boldsymbol{\theta}^*))} d\boldsymbol{\theta}^* + \int_{\boldsymbol{\theta}^* \in \Theta^c} q(\boldsymbol{\theta}^*|\boldsymbol{\phi}) \log(ZC) d\boldsymbol{\theta}^*,
\end{aligned}
\tag{8}
$$

where $Z = \int \exp(-F(\boldsymbol{\theta}^*))d\boldsymbol{\theta}^*$ is the normalization constant. The second term in the above equation equals to $\log(ZC)$, a constant w.r.t. $\boldsymbol{\phi}$, thus could be ignored during optimization.

We then propose our Bayesian loss as:

$$
\begin{aligned}
F_{\mathrm{B}}(\boldsymbol{\phi}) &= \int_{\boldsymbol{\theta}^* \in \Theta^c} q(\boldsymbol{\theta}^*|\boldsymbol{\phi}) \log q(\boldsymbol{\theta}^*|\boldsymbol{\phi})d\boldsymbol{\theta}^* + \int_{\boldsymbol{\theta}^* \in \Theta^c} q(\boldsymbol{\theta}^*|\boldsymbol{\phi})F(\boldsymbol{\theta}^*)d\boldsymbol{\theta}^* \\
&= -H(q(\boldsymbol{\theta}^*|\boldsymbol{\phi})) + E_{q(\boldsymbol{\theta}^*|\boldsymbol{\phi})}[F(\boldsymbol{\theta}^*)],
\end{aligned}
\tag{9}
$$

where the first term of $F_{\mathrm{B}}$ measures the negative entropy of our approximated posterior, and the second term is the expectation of the loss function over of posterior.

**Gaussian Posterior.** For local approximation, we consider $\boldsymbol{\phi} = (\boldsymbol{\mu}, \boldsymbol{\Sigma})$ and $q(\boldsymbol{\theta}^*|\boldsymbol{\phi}) = \mathcal{N}(\boldsymbol{\mu}, \boldsymbol{\Sigma})$, where $\boldsymbol{\mu}$ is the mean vector and $\boldsymbol{\Sigma}$ is the covariance matrix of a normal distribution. For simplicity, we consider $\boldsymbol{\Sigma}$ to be a diagonal matrix: $\boldsymbol{\Sigma} = \mathrm{diag}(\sigma_1^2, \sigma_2^2, \sigma_3^2, ...)$. The second term in Eq (9) involves the integral over $F(\boldsymbol{\theta}^*)$, which is intractable. Therefore, we use Monte Carlo sampling through $q(\boldsymbol{\theta}^*|\boldsymbol{\phi})$ to replace the integral there. However, the direct sampling of the posterior parameters makes it difficult for the optimization as it is inaccessible to get the gradient w.r.t. $\boldsymbol{\mu}$ and $\boldsymbol{\Sigma}$. Moreover, the standard deviation $\sigma_1$, $\sigma_2$, ... must be non-negative, making the optimization constrained.

To overcome those two challenges, we use the trick introduced in (Blundell et al., 2015) to shift sampling from $q(\boldsymbol{\theta}^*|\boldsymbol{\phi})$ to sampling from a standard normal distribution $\mathcal{N}(\boldsymbol{0}, \boldsymbol{I})$. And we reparameterize standard deviation $\sigma_i$ to $\rho_i$ as $\sigma_i = \log(1+\exp(\rho_i))$. Then for any $\boldsymbol{\epsilon}$ sampled from $\mathcal{N}(\boldsymbol{\mu}, \boldsymbol{I})$, we could calculate $\boldsymbol{\theta}^*$ as $\boldsymbol{\theta}^* = \boldsymbol{u} + \log(1+\exp(\boldsymbol{\rho})) \odot \boldsymbol{\epsilon}$, where $\odot$ means element-wise product and $\boldsymbol{\rho} = (\rho_1, \rho_2, ...)$.

## 3.5 BAYESIAN AVERAGING

We recall our goal to build the posterior over the global optimum: $p(\boldsymbol{w}^*|\mathcal{D})$ through Eq 4. We consider using Monte Carlo sampling to approximate the integral as:

$$
p(\boldsymbol{w}^*|\mathcal{D}) = \int p(g(\cdot)|\mathcal{D})p(\boldsymbol{w}^*|\mathcal{D}, g)dg \approx \int_{\boldsymbol{\theta}^* \in \Theta^c} q(\boldsymbol{\theta}^*|\boldsymbol{\phi})p(\boldsymbol{w}^*|g_{\boldsymbol{\theta}^*}(\cdot), \mathcal{D})d\boldsymbol{\theta}^* \approx \sum_{i=1}^{N} p(\boldsymbol{w}^*|g_{\boldsymbol{\theta}_i^*}(\cdot), \mathcal{D})
\tag{10}
$$

where $\boldsymbol{\theta}_i^*$ is sampled from $q(\boldsymbol{\theta}^*|\boldsymbol{\phi})$. Since $q(\boldsymbol{\theta}^*|\boldsymbol{\phi})$ follows a multivariate Gaussian distribution where individual dimensions are independent of each other, we estimate the summation above in

each dimension using independent MC samplings. In practice, $N$ being $10,000$, $100,000$, and $500,000$ led to negligible differences in the 1D estimations, and $N$ was thus fixed at $10,000$.

## 3.6 META-TRAINING SET

In order to boost the robustness and generalizability of our optimizer posterior $p(g^*|\mathcal{D})$, we consider using an ensemble of objective functions $\mathcal{F} = \{f_i\}_{i=1}^N$. Specifically, we replace the objective function in Eq 3 with $\frac{1}{N}\sum_{i=1}^N \sum_{t=1}^T f_i(\boldsymbol{w}_{\boldsymbol{\theta}^*,i}^t)$ and rewrite $F(\boldsymbol{\theta}^*)$ as:

$$F(\boldsymbol{\theta}^*) = \frac{1}{N}\sum_{i=1}^N \sum_{t=1}^T f_i(\boldsymbol{w}_{\boldsymbol{\theta}^*,i}^t) + \lambda||\boldsymbol{\theta}^*||_2 \tag{11}$$

where $\boldsymbol{w}_{\boldsymbol{\theta}^*,i}^t$ is the solution at $t$th iteration for objective $f_i$ optimized by $g_{\boldsymbol{\theta}^*}$. Such replacement can let our posterior generalize to novel objective functions. We regard the functional dataset $\mathcal{F}$ as the meta-training set. During the experiments, we create different meta-training sets for different problems, which will be described in details in Sec 4. We note that Eq 11 is also the objective or part of the objective in many meta-optimizers (Andrychowicz et al., 2016; Chen et al., 2017; Lv et al., 2017; Cao et al., 2019a). However, those methods are focusing on training a deterministic optimizer without uncertainty-awareness.

## 3.7 TWO-STAGE TRAINING FOR EMPOWERING THE LOCAL POSTERIOR

As mentioned before, due to the extreme large optimizer space, we focus on modelling the posterior locally around $\boldsymbol{\theta}^c$, an optimum of interest. If we directly train our model through the Bayesian loss in Eq 9, we simply regard that our posterior is locally around the random initialized point. In order to obtain an real optimum of interest $\boldsymbol{\theta}^c$, we first train our model in a non-Bayesian way through minimizing the loss in Eq 11. We then use $\boldsymbol{\theta}^c$ as the warm start for $\boldsymbol{\mu}$, and start the second Bayesian training stage through the loss in Eq 9. Both training stages are critical for empowering our local posterior. Such statement can be demonstrated through the ablation study in Appendix F.

## 3.8 MODEL ARCHITECTURE, IMPLEMENTATION AND COMPUTATIONAL COMPLEXITY

The model is implemented in Tensorflow 1.13 (Abadi et al., 2016) and optimized by Adam (Kingma & Ba, 2014). For the optimizer architecture, we use the coordinate-wise LSTM from Andrychowicz et al. (2016). We also validate this design choice in Appendix F. Due to the coordinate-wise nature, our BL2O model only contains 10,282 free parameters. For all experiments, the length of LSTM is set to be 20. Both training stages include 5,000 training epochs.

The time complexity for BL2O is $O(KBN_e + KN_eH^2)$, where $K$ is the number of sampling trajectories, $B$ is the minibatch size, $N_e$ is the number of objective parameters, and $H$ is the hidden size of LSTM ($H = 20$ in the study). As the batch size increases, the computational cost is close to the traditional Bayesian neural networks trained through SGD. Due to the coordinate-wise LSTM, the space coomplexity (memory cost) of BL2O is only $O(H^2)$, which remains the same as the number of objective parameters varies. Both the time and the space complexity of BL2O are the same as DM_LSTM (Abadi et al., 2016), while those of Adam are $O(KBN_e)$ and $O(N_e)$, respectively.

## 4 EXPERIMENTS

We test our BL2O model extensively on optimizing: non-convex test functions, energy functions in protein-protein interactions, loss functions in image classification and loss functions in data privacy attack. We compare BL2O to three non-Bayesian methods: Adam, Particle Swarm Optimization (PSO) (Kennedy & Eberhart, 1995), DM_LSTM (Andrychowicz et al., 2016) and a recently published Bayesian method, BAL (Cao & Shen, 2020). All algorithms are running for 10,000 times with random initializing points to obtain the empirical posterior distributions. During each run, the hyperparameters in Adam and PSO are sampled from Table 4 in Appendix A. Out of 10,000 solutions we choose the one with the lowest function value to be the final solution ($\hat{\boldsymbol{w}}$).

Generally, for optimization performance, we assess the distance between the final solution and the global optima: $||\hat{\boldsymbol{w}} - \boldsymbol{w}^*||$. The lower the distance is, the better the solution quality is. For uncertainty quantification, we assess the upper bound $r_\sigma$ and the real confidence $\epsilon_\sigma$ given a fixed

confidence level $\sigma$. The real confidence $\epsilon_\sigma$ is defined by the fraction of 10,000 solutions that actually fall in the bounded region. The lower the $r_\sigma$ is, the tighter the confidence interval is. And the closer of $\epsilon_\sigma$ to $\sigma$ is, the more accurate the confidence estimate is.

**Comparison in optimizing test functions.** We first test the performance on test functions in the global optimization benchmark set (Jamil & Yang, 2013). We choose three extremely rugged, non-convex functions: Rastrigin, Ackley and Griewank in 5 dimensions: 6D, 12D, 18D, 24D, 30D. For each function, we create a diverse, broad family of similar functions $f_j(\boldsymbol{w})$ as the meta-training set used for training DM_LSTM and BL2O. The analytical forms and the meta-training sets of those functions are shown in Table 5 in Appendix B.

We compare BL2O with all 4 competing methods. The optimization and UQ performances are shown in Fig. 1. In all three cases and 5 dimensions, BL2O has led to the best solution quality. In terms of UQ, BL2O has shown the most accurate confidence estimation ($\epsilon_\sigma \approx \sigma$) when $\sigma = 0.9$ and $\sigma = 0.8$, while BAL was the second best. And BL2O has shown much tighter confidence intervals $r_\sigma$ against BAL. In some cases, although DM_LSTM has lower $r_\sigma$ than BL2O, it has much lower confidence level, indicating that this tight upper bound in DM_LSTM is miscalibrated. As a result, BL2O has shown the best performance in both optimization and UQ.

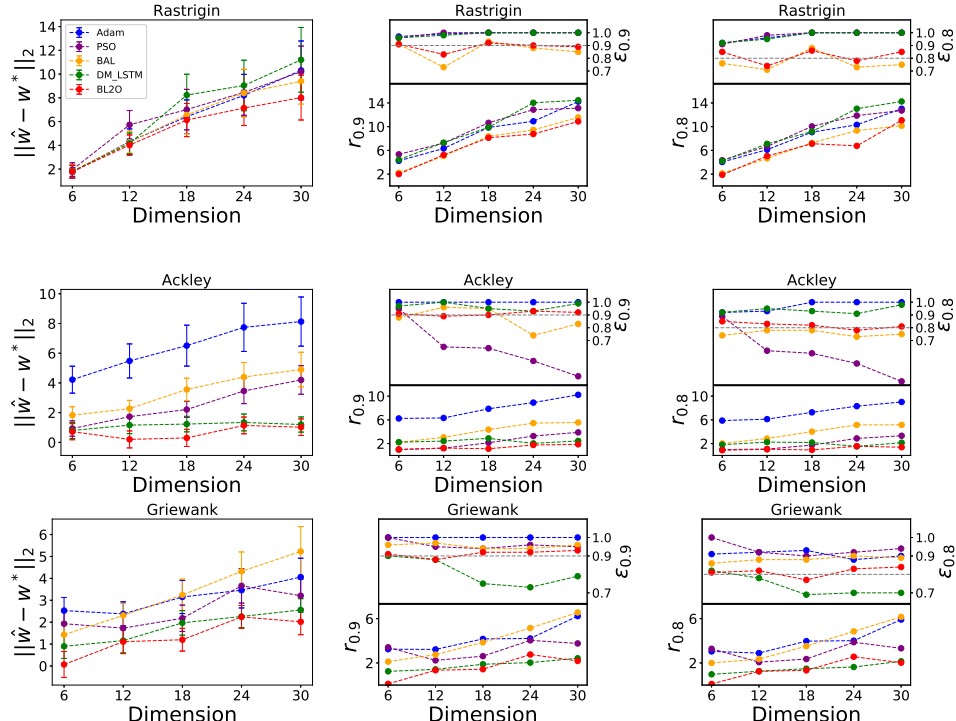

Figure 1: The optimization performance (left) and the UQ performance ($r_\sigma$ and $\epsilon_\sigma$) of different methods on three test functions.

**Comparison in optimizing energy functions for protein docking.** We then apply BL2O to a bioinformatics application: predicting the 3D structures of protein-complexes (Smith & Sternberg, 2002), called protein docking. *Ab initio* protein docking can be recast as optimizing a noisy and expensive energy function in a high-dimensional conformational space (Cao & Shen, 2020): $\boldsymbol{x}^* = \arg\min_{\boldsymbol{x}} f(\boldsymbol{x})$. While solving such optimization problems still remains difficult, quantifying the uncertainty of resulting optima (docking solutions) is even more challenging. In this section, we apply our BL2O to optimization and uncertainty quantification in protein docking and compare with a state-of-the-art method BAL (Cao & Shen, 2020).

We describe the detailed settings of BL2O on protein docking in Appendix C. From BL2O, we obtain a posterior distribution $p(\boldsymbol{w}^*|D)$ over the native structure $\boldsymbol{w}^*$ and the lowest energy structure, $\hat{\boldsymbol{w}}$. In protein docking, the quality of a predicted structure is based on the distance to the native structure (the global optimum): $||\hat{\boldsymbol{w}} - \boldsymbol{w}^*||$. For UQ, we assess the two-sided confidence interval at $\sigma = 0.9$ as $P(l_{0.9} \leqslant ||\hat{\boldsymbol{w}} - \boldsymbol{w}^*|| \leqslant r_{0.9}) = 0.9$.

In Table 1, we assess $||\hat{\boldsymbol{w}} - \boldsymbol{w}^*||$, $r_{0.9} - l_{0.9}$ and whether $||\hat{\boldsymbol{w}} - \boldsymbol{w}^*||$ is within the confidence interval. For optimization, BL2O clearly outperforms BAL in two medium cases while performing slightly worse in the other cases. Yet for UQ, BL2O shows clearly superior performance over BAL in all cases, with accurate or/and tight confidence intervals. We also visulize the posterior distributions over $||\hat{\boldsymbol{w}} - \boldsymbol{w}^*||$ for protein 1JMO_4. As shown in Fig 2, we can see compared to that of BAL, BL2O's distribution has real $||\hat{\boldsymbol{w}} - \boldsymbol{w}^*||$ within the 90% C.I. and smaller variance. More posterior distributions are shown in Appendix D.

Table 1: Performances in optimization and uncertainty quantification on 5 docking cases.

| | $||\hat{\boldsymbol{w}} - \boldsymbol{w}^*||$ | | $r_{0.9} - l_{0.9}$ (Å) | | $||\hat{\boldsymbol{w}} - \boldsymbol{w}^*|| \in [l_{0.9}, r_{0.9}]$? | |
|---|---|---|---|---|---|---|
| Target (docking diffficulty) | BAL | BL2O | BAL | BL2O | BAL | BL2O |
| 1AHW_3 (easy) | 1.89 | 2.07 | 2.20 | 1.98 | No | No |
| 1AK4_7 (easy) | 2.45 | 2.70 | 1.93 | 1.66 | Yes | Yes |
| 3CPH_7 (medium) | 3.89 | 3.21 | 1.70 | 2.20 | No | Yes |
| 1HE8_3 (medium) | 3.05 | 2.32 | 2.24 | 1.61 | Yes | Yes |
| 1JMO_4 (difficult) | 1.45 | 1.55 | 2.90 | 1.26 | No | Yes |

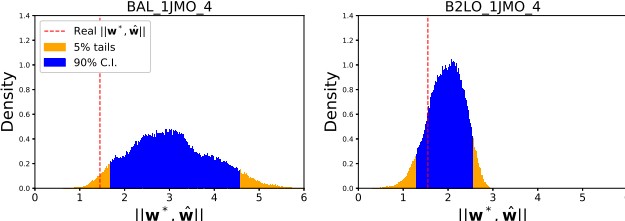

Figure 2: Visualizations of estimated posterior distributions and confidence intervals.

**Comparison in optimizing loss functions in image classification.** We then test the performance of optimizing the loss function in image classification on the MNIST dataset. We apply a 2-layers MLP network as the classifier. The competing methods include Adam, DM_LSTM and two Bayesian neural network methods: variational inference (VI) (Blundell et al., 2015) and Learnable Bernoulli Dropout (LBO) (Boluki et al., 2020). Moreover, for DM_LSTM and BL2O, we apply a trick during the optimizer training called curriculum learning (CL) and introduce it in detail in Appendix E for training over long-term iterations. We call DM_LSTM with CL as DM_LSTM_C and BL2O with CL as BL2O_C.

The assessment of the optimization and UQ for this machine learning task is different from that for optimization before. In terms of optimization, we assess the classification accuracy on the test set. In terms of UQ, we measure two metrics that assess the robustness and trustworthiness of the classifier: the in-domain calibration error and the out-of-domain detection rate.

We first compare the accuracy on the testing set among different methods. As shown in Table 2, Adam, DM_LSTM_C and BL2O_C have almost the same best performance. The significant improvement from DM_LSTM to DM_LSTM_C, and from BL2O to BL2O_C shows the big advantage of curriculum learning in learning to optimize. In conclusion, BL2O_C had on par accuracy with Adam and DM_LSTM_C on the MNIST dataset.

However, classification models must not only be accurate, but also indicate when they are likely to be incorrect. Confidence calibration, the probability that estimates the true likelihood of each prediction is also important for classification models. In the ideal case, the maximum output probability (MaxConfidence) for each test sample should be equal to the prediction accuracy for that sample. To assess the calibration of each methods, we split the test set into 20 equal-sized bins and assess the **calibration error** as the average discrepancy between accuracy and MaxConfidence in each bin. As seen in Table 2, among all methods compared, BL2O_C and BL2O had the least calibration error. The figure of Acc. vs MaxConf. is also shown in Fig. 4 in Appendix E.

We also inspect the out-of-domain detection of BL2O, BL2O_C and competing methods. We train all models on the data belonging to the first 5 classes in the MNIST training dataset (the last layer of the optimizee is modified to have 5 rather than 10 neurons) and test them on the remaining

Table 2: Performance of classification on the MNIST test set.

| Models | In-Domain Accuracy (%) | In-Domain Calibration Error | Out-of-Domain Detection | |
| --- | --- | --- | --- | --- |
| | | | $q_{.4}(\%)$ | $q_{.5}(\%)$ |
| Adam | 93.2 | 5.0E-4 | 0.7 | 2.8 |
| DM_LSTM | 81.0 | 4.2E-3 | 0.4 | 2.0 |
| DM_LSTM_C | 93.4 | 9.9E-4 | 4.6 | 10.6 |
| VI | 87.8 | 4.4E-3 | 4.8 | 10.6 |
| LBO | 91.2 | 9.6E-4 | 5.2 | 11.3 |
| BL2O | 90.1 | 4.9E-4 | **29.9** | **31.8** |
| BL2O_C | **93.5** | **4.3E-4** | 12.4 | 20.9 |

samples from the other 5 classes. An ideal model would predict a uniform distribution over the 5 wrong classes. Therefore, we define the **out-of-domain detection rate** at threshold $t$, $q_t$, as the percentage of test samples with max class confidence below $t$. The larger the $q_t$, the better out-of-domain detection is. As shown in Table 2, BL2O and BL2O_C shows superior performance with all competing methods. Notably, BL2O without curriculum learning had much better out-of-domain detection rates compared to BL2O with curriculum learning.

**Comparison in optimizing loss functions for data privacy attack.** We finally apply our model to an application that critically needs UQ. As many machine learning models are deployed publicly, it is important to avoid leaking private sensitive information, such as financial data, health data and so on. Data privacy attack (Nasr et al., 2018) studies this problem by playing the role of hackers and attacking the machine-learning models to quantify the risk of privacy leakage. Better attacks would help models to be better prepared for privacy defense.

We use the model and dataset in (Cao et al., 2019b), where each input has 9 features involving patient genetic information and the output $p$ is the probability of the clinical significance (having cancer or not) for a patient. We study the following model inversion attack (Fredrikson et al., 2015): by giving 5 features $\boldsymbol{w}' \in [0,1]^5$ out of 9 and the label $p$ of each patient, we want to recover the rest 4 features $\boldsymbol{w}^* \in [0,1]^4$ (potentially sensitive patient information). Therefore, for each patient, the objective is $\boldsymbol{w}^* = \arg\min_{\boldsymbol{w} \in [0,1]^4} (m(\boldsymbol{w}', \boldsymbol{w}) - p)^2$, where $\boldsymbol{w}^*$ is the ground-truth of $\boldsymbol{w}$ and $m$ is the trained predictive model. The closeness between the predicted and the real input features can quantify the risk of information leakage and the quality of the attack. We compare BL2O with Adam, PSO, BAL and DM_LSTM on optimization and UQ on all test cases in (Cao et al., 2019b). The meta-training objectives for BL2O and DM_LSTM are the training set in (Cao et al., 2019b).

As shown in Table 3, BL2O has shown the best performance in both optimization and UQ compared to all competing methods. It is noteworthy that learned optimizers (DM_LSTM and BL2O) had much better optimization performance than pre-defined optimizers. And the Bayesian methods (BAL and BL2O) had significantly better UQ performance than non-Bayesian methods. BL2O possessed the advantages of both learned and Bayesian optimizers to achieve the best performance.

Table 3: The optimization and UQ performance of different methods on data privacy attack.

| | $\|\|\hat{\boldsymbol{w}} - \boldsymbol{w}^*\|\|$ | $r_{0.9}$ | $\|\epsilon_{0.9} - 0.9\|$ | $r_{0.8}$ | $\|\epsilon_{0.8} - 0.8\|$ |
| --- | --- | --- | --- | --- | --- |
| Adam | 0.45 | 0.74 | 0.10 | 0.63 | 0.20 |
| PSO | 0.32 | 0.82 | 0.10 | 0.72 | 0.20 |
| BAL | 0.34 | 0.53 | 0.06 | 0.41 | 0.08 |
| DM_LSTM | 0.20 | 0.52 | 0.10 | 0.47 | 0.19 |
| BL2O | **0.17** | **0.43** | **0.05** | **0.32** | **0.04** |

## 5 CONCLUSION

Current optimization algorithms, even with uncertainty-awareness, do not address the uncertainty arising within the optimizer itself. To close this gap, we parameterize the update rule as a neural network and build a Boltzmann-shaped posterior over the algorithmic space. We apply our Bayesian Learning-to-Optimize (BL2O) framework to optimize test functions, energy functions in protein docking, loss functions in image classification and loss functions in data privacy attack. The empirical results demonstrate that BL2O outperforms the state-of-the-art methods in both optimization and uncertainty quantification, as well as the calibration and out-of-domain detection in classification.

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

# A OPTIMIZER DISTRIBUTION SETTINGS FOR ADAM AND PSO

| Methods | Optimizer Distribution Settings |
|---|---|
| Adam | $\log_{10}(\text{lr}) \sim \text{U}[\text{-2, -1}], \beta_1 \sim U[0.9, 1.0], \beta_2 \sim U[0.999, 1.0]$ |
| PSO | $w \sim \text{U}[0.5, 1.5], \text{C1} \sim \text{U}[1.5, 2.5], \text{C2} \sim \text{U}[1.5, 2.5]$ |

Table 4: The optimizer distributions over hyperparameters in Adam and PSO.

# B ANALYTIC FORMS AND META-TRAINING SETS OF TEST FUNCTIONS

| Function Name | Analytic form |
|---|---|
| Rastrigin | $f_j(\boldsymbol{w}) = \|\|\boldsymbol{w}\|\|_2^2 - \sum_{i=1}^{n} 10\cos(2\pi w_i) + 10n$ |
| Ackley | $f_j(\boldsymbol{w}) = -20\exp(-0.2\sqrt{0.5\|\|\boldsymbol{w}\|\|_2^2}) - \sum_{i=1}^{n}\exp(\cos(2\pi w_i)/n) + e + 20$ |
| Griewank | $f_j(\boldsymbol{w}) = 1 + \frac{1}{4000}\sum_{i=1}^{n}\|\|\boldsymbol{w}\|\|_2^2 - \prod_{i=1}^{n}\cos(w_i)$ |
| | Meta-training set |
| Rastrigin | $f_j(\boldsymbol{w}) = \|\|\boldsymbol{A}_j\boldsymbol{w} - \boldsymbol{b}_j\|\|_2^2 - 10\boldsymbol{c}_j\cos(2\pi\boldsymbol{w}) + 10n$ |
| Ackley | $f_j(\boldsymbol{w}) = -20\exp(-0.2\sqrt{0.5\|\|\boldsymbol{A}_j\boldsymbol{w} - \boldsymbol{b}_j\|\|_2^2}) - \exp(\boldsymbol{c}_j\cos(2\pi\boldsymbol{w})/n)$ |
| Griewank | $f_j(\boldsymbol{w}) = 1 + \frac{1}{4000}\sum_{i=1}^{n}\|\|\boldsymbol{A}_j\boldsymbol{w} - \boldsymbol{b}_j\|\|_2^2 - \prod_{i=1}^{n}(\cos(w_i) + c_{ji} - 1)$ |

Table 5: The analytic forms of test functions and the meta-training sets used for training DM_LSTM and BL2O, where n is the dimension and $\boldsymbol{A}_j \in \mathbb{R}^{n \times n}$, $\boldsymbol{b}_j \in \mathbb{R}^{n \times 1}$ and $\boldsymbol{c}_j \in \mathbb{R}^{n \times 1}$ are parameters whose elements are sampled from i.i.d. normal distributions. It is obvious that the test functions are the special cases in the training sets, respectively.

## C  SETTINGS FOR PROTEIN DOCKING EXPERIMENTS

We calculate the energy function (objective function $f(\boldsymbol{x})$) in a CHARMM 19 force field as in (Moal & Bates, 2010). 25 protein-protein complexes are chosen from the protein docking benchmark set 4.0 (Hwang et al., 2010) as the training set, which is shown in Table 6. For each target, we choose 5 starting points (top-5 models from ZDOCK (Pierce et al., 2014)). In total, our training set includes 125 samples. Moreover, we parameterize the search space as $\mathbb{R}^{12}$ as in BAL (Cao & Shen, 2020). The resulting $f(\boldsymbol{x})$ is fully differentiable in the search space. We only consider 100 interface atoms due to the computational concern. The number of iterations for one training epoch is 600 and in total we have 5000 training epochs. Both BL2O and BAL have 600 iterations during the testing stage. For fair comparison, after optimization, we rescore the BL2O samples for UQ using the scoring function (random forest) in BAL.

Table 6: 4-letter ID of proteins used in the training set.

| Difficulty level | Protein Data Bank (PDB) code |
|---|---|
| Rigid | 1N8O , 7CEI , 1DFJ , 1AVX , 1BVN , 1IQD , 1CGI , 1MAH , 1EZU , 1JPS , 1PPE , 1R0R , 2I25 , 2B42 , 1EAW , 2JEL , 1BJ1 , 1KXQ , 1EWY |
| Medium | 1XQS , 1M10, 1IJK ,1GRN |
| Flexible | 1IBR , 1ATN |

# D  POSTERIOR DISTRIBUTIONS IN PROTEIN DOCKING

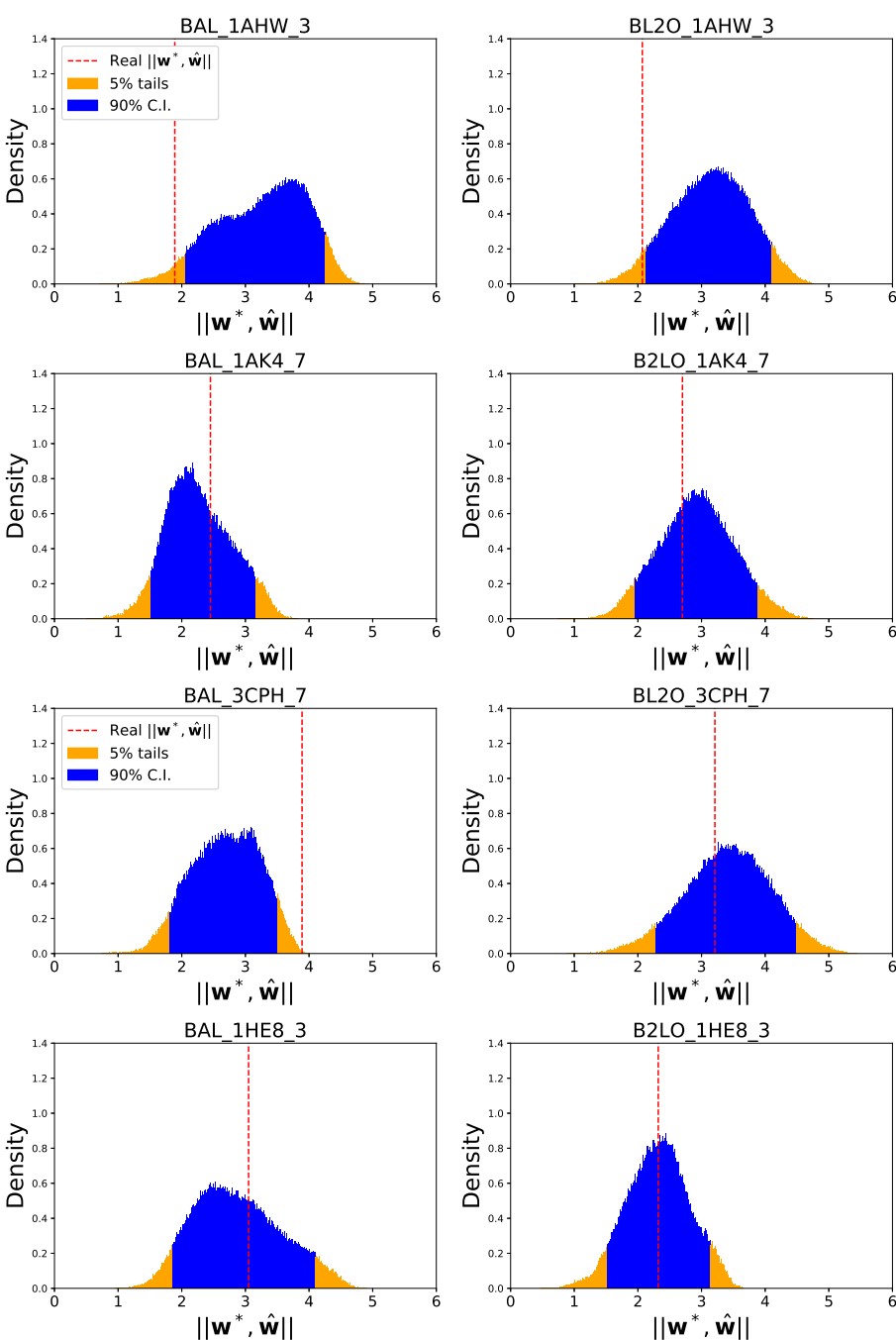

Figure 3: Visualizations of estimated posterior distributions and confidence intervals for more docking cases.

# E  CURRICULUM LEARNING AND CALIBRATION FIGURE IN IMAGE CLASSIFICATION

A common issue in learning to optimize for neural network training is that: the optimizer training usually takes hundreds of iterations, while training a neural network usually costs thousands or tens of thousand iterations. For instance, for a MNIST training dataset consisting of 50,000 images, training a neural network for 100 epochs corresponds to almost 20,000 iterations with a batch size of 128.

Hundreds of iterations are good for the first few epochs during optimizer training, since we would like to only focus on decreasing the loss in the first hundreds of iterations. However, as training goes on, we would like to see that the later iterations could also decrease the training loss. During this stage, only hundreds of iterations are clearly not enough.

In order to overcome this issue, we bring the idea from (Bengio et al., 2009). Specifically, We set a list for the number of iterations: [100, 200, 500, 1000, 1500, 2000, 2500, 3000] and we gradually increase the number of iterations following the list for every 100 epochs in optimizer training if the optimizee loss is decreasing. Once the number reaches 3000, it will not change any more until the training ends.

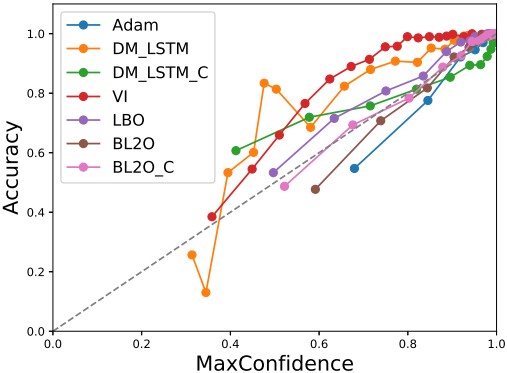

Figure 4: The calibration figure in image classification.

# F  ABLATION STUDY.

In order to validate various design choices, we perform the ablation study as follows:

- **B1**: We use the coordinate-wise gated recurrent unit (GRU) network as the optimizer architecture. We train our model directly on the Bayesian loss: Eq 9 without non-Bayesian training.
- **B2**: We replace the GRU network with the LSTM network.
- **BL2O**: We add the non-Bayesian training stage to find a local optimum of interest first before training on the Bayesian loss.

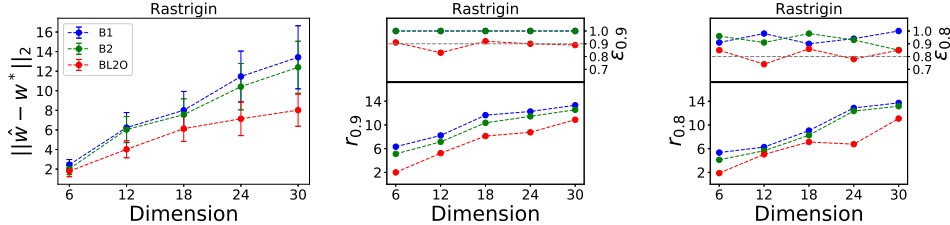

Figure 5: Ablation study: the optimization performance (left) and the UQ performance ($r_\sigma$ and $\epsilon_\sigma$) of different models on the Rastrigin function.

We test these three models on the Rastrigin test function. As shown in Fig 5, **B2**(LSTM) has slightly better performance in both optimization and UQ compared to **GRU**. But BL2O has shown superior performance compared to both **B1** and **B2** in optimization and UQ. These results clearly demonstrate that the two training stages must be coupled together to empower the local posterior.

