# OpenReview forum: "Bayesian Learning to Optimize: Quantifying the Optimizer Uncertainty"
_ICLR.cc/2021/Conference — Reject_

### Official Review · AnonReviewer3 · 2020-10-26
**Novel but Has Important Weaknesses**

**Rating:** 4
**Confidence:** 4

**Review:**

Summary:
This paper proposes to account for an additional source of uncertainty during inference, i.e., the uncertainty introduced by the optimization algorithm. The paper uses a neural network to parameterize the optimization algorithm, and proposes intuitive and heuristic techniques for inferring its posterior distribution.

Strong points:
- The paper provides a novel perspective by considering the uncertainty introduced by the optimization algorithm, which is novel as far as I know.
- The experiments cover a diverse set of applications, which indeed showcases the generality of the proposed algorithm.

Weak points:
- Since the main contribution of the paper is accounting for the uncertainty introduced by the optimization algorithm which is very novel, I think a more rigorous definition of this source of uncertainty is needed. In particular, I think the paper needs a clear definition and illustration of $p(g(\cdot)|\mathcal{D})$ which is the ultimate distribution the paper is trying to approximate. Analogously, in standard posterior inference, $p(\omega|\mathcal{D})$ is defined through the prior $p(\omega)$ and likelihood $p(\mathcal{D}|\omega)$ which links the observed data $\mathcal{D}$ and the model parameter $\omega$ by defining how likely $\mathcal{D}$ is generated by a given $\omega$. However, I don't see such a clear link if I replace $\omega$ by $g(\cdot)$: how should the likelihood $p(\mathcal{D}|g(\cdot))$ be interpreted? Does this mean the choice of the optimization algorithm affect how the observed data is generated?
- Section 2, first paragraph: it says here that for these Bayesian optimization algorithms, "significant approximation is needed". But I wonder whether this can be framed as a disadvantage of these algorithms? I think it should not be a disadvantage as long as these approximations deliver good performances. Also it would be interesting to see the proposed method applied to Bayesian optimization, although this is not an important point.
- The paper measures the quality of the solution $\hat{\omega}$ by $||\hat{\omega}-\omega^*||$ (Equation 3 and in the experimental results). However, I think this may not be very appropriate and $||f(\hat{\omega})-f(\omega^*)||$ would be a better measure, because if $f$ is very non-smooth, even if $||\hat{\omega}-\omega^*||$ is small, $f(\hat{\omega})$ may be very large.
- I think many design choices of the algorithm are too heuristic and lack a rigorous justification. For example, at the bottom of page 3, it is claimed that "it is important to optimize the following loss" of Equation 5, but why? Also, defining the different objective functions $f_j(\cdot)$ by different mini-batches (Equation 6) seems like a convenient and arbitrary choice, and doesn't seem very rigorous. This further calls into question the reliability of the experimental comparisons, since it seems that there isn't a rigorous way to choose these objective functions (I think referred to as "meta-training set/objectives" in the Experiments section), and different choices may affect the performance (this should be explored empirically if possible).
- For experiments, most baselines under comparison are non-Bayesian methods. This doesn't seem very appropriate since the goal of the paper is to learn a better $p(\omega^*|\mathcal{D})$ (Equation 4) which is a distribution. I think more Bayesian methods than non-Bayesian methods should have been compared with. For example, for the image classification experiment, the only Bayesian method under comparison is the VI paper from 2015, since then, there have been a number of more recent works on Bayesian neural networks, and it would be better if more comparisons with BNN is performed.
- In the experimental comparisons, the paper claims that a tighter confidence bound $r_{\sigma}$ is better. However, I think this may not be the case. For example, if we are training a Bayesian neural network using a very small dataset, then the posterior $p(\omega^*|\mathcal{D})$ isn't supposed to be concentrated right?
- Table 2: It seems without the curriculum learning trick (which is not a main contribution of this paper), BL2O is significantly outperformed by Adam.

More minor points:
- Section 2, first paragraph, third line: here "Bayesian optimization" should be replaced by some specific instances of Bayesian optimization algorithms such as predictive entropy search, since what's described here isn't how a generic BO algorithm works but only some BO algorithms. It could be misleading to some readers not familiar with BO.
- Section 2, last paragraph: I think this paragraph (maybe except for the first sentence) should belong to the end of Introduction instead of here.
- Figure 1: Error bars should be included.


----------------------Update After Rebuttal----------------------------------

I appreciate the response from the authors, and the authors' efforts in significantly revising the Method section. The Method section in the current version looks much better and clear. Most points from the authors' response are reasonable, and some of my concerns are indeed cleared, such as my questions regarding the use of $||\hat{\omega}-\omega^*||$ instead of $||f(\hat{\omega})-f(\omega^*)||$. However, in the revised paper, I still find a few places questionable, so I still have a few concerns which are still about the first and fourth points I raised in my original review:

- In Definition 3.2 of the revised paper, I find this definition of optimal optimizer $g^*$ not fully convincing. I understand that this definition of $g^*$ naturally gives rise to the $F(\theta^*)$ as in the first line of page 5, however, since our problem is an optimization problem (i.e., to minimize $f$), I think the summation in the definition of $g^*$ should be replaced by minimization. The current definition of using a summation over different iterations could lead to problems in some scenarios. For example, imagine we have optimizer B who quickly converges to a local minimum, and optimizer B who explores the entire search space first (encountering many large $f$ values along the way) and finally converges to the global minimum. Then according to this definition, optimizer A is likely to be defined to be better than optimizer B, which is incorrect. Moreover, another problem with the current definition is that the initialization $\omega^1_g$ is not specified. I feel that for the sake of defining the optimal optimizer, the argmin over all optimizers $g\in\mathcal{G}$ should be based on the same initialization for all optimizers, i.e., $\omega^1_g$ is the same for all $g\in\mathcal{G}$. Since I imagine that for different initializations, the optimal optimizer could be different.

- Equation (5) on page 4, I think the distribution of the initial point $p(\omega^1_{\theta^*})$ should appear on the Right Hand Side. Because given the optimizer $\theta^*$, the distribution of the trajectory $\mathcal{D}$ clearly depends on the initialization. This may not be a serious problem since $p(\omega^1_{\theta^*})$ could be factored into the normalization constant of $p(\theta^*|\mathcal{D})$.

- Section 3.6, I still find the motivation for using the Meta-Training Set heuristic. It is stated here that the meta-training set is introduced here to improve the robustness and generalizability of solutions, which seems unrelated to the main objective of quantifying optimizer uncertainty. I think (not sure though) a better motivation could be related to uncertainty regarding the function $f$.

- Although the objective of the paper is to "further" consider the uncertainty regarding the optimizer, I feel that the introduced method ONLY considers optimizer uncertainty, and hasn't dealt with $p(\omega^*|\mathcal{D},g)$ in a rigorous way. If I understand correctly, samples from $p(\omega^*|\mathcal{D},g)$ are obtained by running optimizer $g$ for multiple random initialization points (line 5 of Section 4), and I find this kind of heuristic.

Overall I find this paper very interesting mainly due to the novel perspective of considering this additional source of uncertainty, so although I cannot recommend for acceptance this time, I believe it will be a valuable contribution to the community if the problem formulation can be made more rigorous.

---

> ### Author Response · Authors · 2020-11-23
> **Responses to AnonReviewer3**
>
> We thank the reviewer for recognizing the significance and the novelty of our work.  We have addressed the comments as follows and accordingly substantially revised the manuscript, especially the Method section.
>
> **Needs Rigorous Definition of Optimizer Uncertainty**
>
> We agree with the reviewer and have now revised the Method section to 1) use Sec. 3.1 to define the problem of optimization and UQ and motivate the need of optimizer uncertainty; 2) use Sec. 3.2 to formally define the optimizer space $\mathcal{G}$, the unknown optimal optimizer $g^*$ as a random vector in the space, and the optimizer uncertainty.  Specifically, for optimizer uncertainty, we have introduced the prior $p(g*)$, the likelihood $p(\mathcal{D}|g*)$, and the posterior $p(g*|\mathcal{D})$.
>
> As to how the likelihood $p(\mathcal{D}|g*)$ should be interpreted, indeed, the observed data $\mathcal{D}$ are samples during optimization (for instance, trajectories of sample coordinates and gradients when it comes to iterative first-order optimizers) and are thus dependent on the optimizer.
>
> **Expression about Bayesian optimization**
>
> We agree that the claim about the approximation for Bayesian optimization is not necessarily always a disadvantage toward optimization and UQ performances.  We thank the reviewer and have removed the unsubstantiated statement.
>
> **Concern about Using $||\hat{\pmb{w}}-\pmb{w}^*||$ instead of $||f(\hat{\pmb{w}})-f(\pmb{w}^*)||$.**
>
>  We thank the reviewer for raising this concern. We respectively disagree with this point. In many optimization applications, the objective function actually plays the role of the ‘tool’ instead of  the ‘goal’.  That means the objective function value is not the key metric to assess the quality of the solution, which is particularly true for nonconvex and noisy objective functions.   For instance, in protein structure prediction, the goal is to find the native structure located at the global minimum of  the energy functions. For an estimated structure, the quality is measured by the ‘distance’ between this structure to the native structure whereas the energy proximity provides no clue to the model utility.  Also, in the data privacy attack,  the goal is to find the real input  feature vector $\pmb{w}^*$ by optimizing the loss function $f(\pmb{w})$. The solution quality is measured by $||\hat{\pmb{w}}-\pmb{w}^*||$.  The way for measuring the solution quality in those applications  actually motivates our choice for $||\hat{\pmb{w}}-\pmb{w}^*||$ as the assessment metric.
>
> **Design Choice for the Algorithm Seemed Heuristic**
>
> We apologize that our presentation made the choice of objective functions appear heuristic.  We have completely re-structured and revised the contents. Before formally defining the meta loss in Sec. 3.6,  we now use Sec. 3.4-3.5 to derive the meta loss with justification.  We note that the meta loss form shows the first glimpse in Eq. (7) when the prior $p(g^*)$ is Gaussian and the posterior $p(g^*|\mathcal{D})$ is Boltzmann-shaped.
>
> **Comparison to More Bayesian Methods**
>
> We appreciate the suggestion and have now included one of the latest Bayesian optimization methods in our computer vision experiments: Learnable Bernoulli Dropout (LBO) (Boluki 2020).  The conclusion stays that BL2O had close-to-the-best in-domain accuracy and calibration, while far outperforming all other methods in out of domain (OOD) detection.
>
> **Clarification about the Tighter Confidence Bound**
>
> We think this confusion comes from our presentation, for which we apologize.  We are not using the tightness of confidence interval alone as the only assessment metric regardless of the specific problem and do not compare them across different problems.  Rather, for a given problem (such as a very small dataset), we would like to have a confidence interval as tight as possible, when the confidence level is accurate enough.
>
> **BL2O without the Curriculum Learning Trick Outperformed by Adam?**
>
> We would like to bring to the reviewer's attention that the statement was not the full story.  Although BL2O without curriculum learning was slightly worse than Adam in in-domain accuracy (90.1\% vs. 93.2\%), it edged Adam to achieve lower in-domain calibration error (4.3E-4 vs. 5.0E-4).  More importantly, it far outperformed Adam in out-of-domain detection (29.9\% vs. 0.7\% in $q_{.4}$) and even outperformed BL2O with curriculum learning whose $q_{.4}$ was 12.4\%.
>
> **Minor points**
>
> We thank the reviewer for the thorough review!  We have adopted the suggestions as follows: 1) We have revised the first paragraph of Sec. 2 to clarify that the statement was about a few cited methods rather than BO in general; 2) We have moved the last paragraph of Sec. 2 (except the first sentence) to the end of Sec. 1 and revised the contents according to the changes in the Methods section; and 3) We have now included error bars in Fig. 1 (optimization performance in the left panels).

---

### Official Review · AnonReviewer4 · 2020-10-28
**Understanding the effect of optimizer uncertainty**

**Rating:** 6
**Confidence:** 3

**Review:**

**(Summary)**
In order to model the uncertainty of optimizers the authors model the algorithmic space of optimizers by a neural network and use a Boltzmann-shaped posterior posterior over this space.  The use VI to approximate the intractable posterior distribution. Finally, they show that they demonstrate their approach on a diverse set of benchmarks.

**(Strong points)**
- This is an interesting problem and an interesting approach to understanding the effect of optimizer uncertainty.
- The experimental section is strong and demonstrates that the proposed method works on a large variety of problems.

**(Weak points)**
- The paper is quite hard to read due to a rather large number of grammatical errors and typos. The authors need to spend some more time working on the text to make the paper easier to read.
- I would have appreciated a more thorough discussion on why modeling the space of algorithms by a neural network is a good idea as there are many other possible modeling options that aren’t discussed.

**(Recommendation)**
- I think this paper may be suitable for publication if the authors improve the quality of the text and address my questions in this rebuttal.

**(Questions for the authors)**
- Why do you choose to look at $||\hat{w} - w^*||$ instead of $|f(\hat{w}) - f(w^*)|$? To what extent does noise in the function observations play into this decision?
- As theta are the parameters of the neural network, the dimensionality of the integral in 3.5 must be quite large and I can imagine that $N=10,000$ is far from enough to get an accurate estimate.  What motivates this choice and did you explore QMC or other variance reduction techniques?
- For appendix A, why isn’t the learning rate for Adam log-uniform as it is usually modeled in a log-scale?
- How well are you able to approximate the intractable posterior?

**(Additional feedback)**
- The narrative is a bit confusing. In section 2 you describe related work that is mostly zeroth order methods and then in section 3 you start talking about first order methods, which are the focus of the paper.
- While many methods can be written as the following weight update rule, there are methods that can’t, such as Bayesian optimization with gradients.

---

> ### Author Response · Authors · 2020-11-23
> **Responses to AnonReviewer4: Part 1**
>
> We thank the reviewer for recognizing the significance and merits of our work. We have addressed all the comments  and substantially updated our manuscript accordingly.  In particular, the Method section is re-structured and revised for a more clear logic flow.
>
> **Why Modeling the Space of Algorithms $\mathcal{G}$ by a Neural Network.**
>
> We are thankful that the reviewer suggested this discussion.  We have now revised Sec. 3.3 to dedicate to the discussion.
>
> First of all, a simple way to parameterize the optimizer space for classical optimizers (e.g. Gradient Descent, Adam) is based on their hyperparameters: $\mathcal{G} = \mathcal{H}$, where $\mathcal{H}$ is the hyperparameter space. For instance, for gradient descent, we have $\mathcal{H}=(\alpha)$, where $\alpha$ is the learning rate. For Adam, we have $\mathcal{H}=(\alpha, \beta_1, \beta_2)$, where $\beta_1$ and $\beta_2$ are the coefficients used for computing running averages of gradient and its square.
>
> However, such parameterization  has significant drawbacks. The resulting algorithmic space $\mathcal{G}$  is very limited and heavily depends on the specific optimizer. The $\mathcal{G}$ (a 1D space) parameterized by the hyperparameters of gradient descent is different from that (a 3D space) parameterized by the hyperparameters of Adam. In fact, each is a rather restricted region of the actual $\mathcal{G}$.  The intrinsic flexibility (uncertainty) that lies in an iterative optimizer's update rule is not explored at all in this parameterization. These drawbacks are empirically demonstrated in Sec. 4.
>
> In order to reasonably and accurately model the intrinsic uncertainty within the update rule, we need to find a much more flexible way for modelling $g$. We thus consider to parameterize the optimizer space as a neural network: $\mathcal{G}= \Theta$, where each $\pmb{\theta} \in \Theta$ are the parameters in the neural network. Overcoming drawbacks of the optimizer space $\mathcal{H}$ by hyperparameters, $\Theta$ by neural network parameters generalizes update rules through neural networks that can represent a wide variety of functions. We note that this is also the space of meta-optimizers that learn to optimize (L2O) iterative update rules from data on a given task  (see andrychowicz2016learning, chen2017learning, lv2017learning, cao2019learning). However, there has been no notion of uncertainty let alone the task of UQ  for the learned optimizer in these L2O methods, which is to be addressed in our Bayesian L2O (BL2O).
>
> **Concerns about Using $||\hat{\pmb{w}}-\pmb{w}^*||$ instead of $||f(\hat{\pmb{w}})-f(\pmb{w}^*)||$.**
>
> We thank the reviewer for asking this question and now clarify this in Sec 3.1.  In many optimization applications, the objective function actually plays the role of the ‘tool’ instead of  the ‘goal’.  That means the objective function value is not the key metric to assess the quality of the solution, which is particularly true when objective functions are nonconvex and noisy.  For instance, in protein structure prediction, the goal is to find the native structure located at the global minimum of  the energy functions. For an estimated structure, the quality is measured by the proximity between the estimated and  the native structure (rather than their energy value difference).   In the data privacy attack,  the goal is to find the real input  feature vector $\pmb{w}^*$ by optimizing the loss function $f(\pmb{w})$. The solution quality is again measured by $||\hat{\pmb{w}}-\pmb{w}^*||$.  The way  for measuring the solution quality in those applications  actually motivates our choice for $||\hat{\pmb{w}}-\pmb{w}^*||$ as the assessment metric for solution quality.
>
> **The accuracy of the estimate in 3.5**
>
> We thank the reviewer for asking this question.  Indeed, the dimensionality of $\pmb{\theta}^*$ can be high when those $\pmb{\theta}$ are parameters of neural networks.  That is exactly why we approximated $q(\pmb{\theta}^*|\pmb{\phi})$ as a multivariate Gaussian vector whose individual components are independent of each other.  So we estimate the summation in 3.5 in each dimension using independent MC samplings.  In practice,  $N$ being $10,000$, $100,000$, and $500,000$ led to negligible differences in the 1D estimations, and $N$ was thus fixed at $10,000$.
>
> The treatment significantly increases the efficiency of the approximation and its accuracy can be further improved.  For instance, in future, we may consider a low rank approximation for the off-diagonal entries of the covariance matrix rather than treating them as zeros.
>
> **Learning rate of Adam**
>
> We apologize for the oversight.  We have now adopted log-uniform sampling for Adam's learning rate and updated the corresponding results.  Two more hyperparameters are additionally considered.  The major conclusions maintain about our BL2O's advantages over Adam.

---

> > ### Author Response · Authors · 2020-11-23
> > **Responses to AnonReviewer4: Part 2**
> >
> > **How well is the posterior approximated**
> >
> > This is a great question.  To make the posterior estimation efficiently tractable, we have made approximations.  First of all, due to the extreme large and rugged posterior landscape, modelling the entire posterior is impossible. Instead, we focus on approximating the posterior around a local optimum derived from non-Bayesian optimization.  Second, even in the local region, due to the high dimension of $\bm{\theta}$, we assumed independence among the high-dimensional multivariate Gaussian vector's components (see more in response part 1 about the accuracy of the estimate).  Although there is clearly much space to improve the accuracy (such as the aforementioned low-rank modeling of off-diagonals for the covariance matrix), we believe that the current version has demonstrated the power of the framework in improving optimization and UQ.
> >
> > **Narratives about first-order methods**
> >
> > We thank the reviewer for pointing this out.  We have revised Sec. 2 to include first-order optimization methods.
> >
> > **The universality of the update rule**
> >
> > We agree with the reviewer and have revised the expression as "Considering that many widely-used optimizers are iterative and using first-order derivatives, we restrict the optimizer space as follows:  ..."

---

### Official Review · AnonReviewer5 · 2020-11-03
**Confusing paper**

**Rating:** 5
**Confidence:** 2

**Review:**

###########
 Summary:

The paper considers the question of quantifying the uncertainty that arises from the optimiser used to perform inference in a given model. Taking a Bayesian approach, the aim is to deduce the posterior over the space of optimisers. The form for the posterior is chosen to be a Boltzmann distribution which is then approximated with a multivariate Gaussian using a KL divergence. The parameterisation for the posterior is defined using an LSTM neural network.


###########
Reasons for score:

The high-level idea in the paper is intriguing but I found the logic in the methodology hard to follow and interpret, and it is not clear to me how this work is able to estimate the uncertainty or why it would improve the performance of existing approaches to optimisation.


##########
Pros:

- The main high-level idea seems clear.
- The experiment section is quite extensive and the results provided suggest the approach may have some merit.


##########
Cons:

- The presentation is not clear, especially in the method section. It is rather strange that instead of defining the prior over possible optimisers, the paper only talks about the posterior.
- It seems to me that this work relies heavily on the work of Ortega et al (2012), and if that is the case, a clearer presentation of the main arguments of that work would be very useful.
- The arguments in the problem statement seem quite scattered. For example, I'm not sure I see the relation of Eq. (3) to the rest of this work. Could you explain this statement from page 3 in some more detail:
"It is straightforward to first model the posterior over the global optima (p(w∗|D)) and then sample from the posterior to obtain..."
- Some ablation studies may clarify the effect of the various design choices. For example, (1) using an LSTM as a way to parameterise the model, (2) the choice of a local region for the local posterior.
- One of your criticisms of MC estimates of optimiser uncertainty is that "it heavily relies on the pre-defined distributions (discrete grids) over the hyperparameters and(or) the start points". However, in your case, the model seems to be constrained to a local neighbourhood of the input parameters, which seems even more limiting to me.
- The overall quality of the writing could be improved (for example, Sec. 1 and 2 include a lot of repetition).

---

> ### Author Response · Authors · 2020-11-23
> **Responses to AnonReviewer5**
>
> We thank the reviewer for the detailed review and important feedback.  Seeing that our presentation has led to confusions, we have addressed all comments including many clarifications.  Furthermore we have revised the ``Methods section for a more clear logic flow.  We would greatly appreciate it if you could kindly re-assess our work after the revised presentations, hopefully more positively.
>
> **Unclear presentation**
> We have now substantially revised the narratives of the Methods section.  After defining the problem of optimization and UQ in Sec. 3.1, we formally define in Sec. 3.2 the optimizer space $\mathcal{G}$ as well as the optimal optimizer $g^*$ as a random vector in $\mathcal{G}$ (including the prior $p(g^*)$, the likelihood $p(\mathcal{D}|g^*)$ and the posterior of the optimizer uncertainty as $p(g^*| \mathcal{D})$).  Based on such definitions, we reveal the computational challenges to model $p(g^*| \mathcal{D})$ and introduce the following contributions to the proposed Bayesian Learning to Optimize (BL2O) framework.
>
> First, to capture the optimizer space $\mathcal{G}$ for a wide class of first-order iterative optimizers and yet to make the calculation of $p(g^*| \mathcal{D})$ tractable, we parameterize $\mathcal{G}$ with parameters $\pmb{\theta}$ of neural networks.  Therefore, $p(g^*|\mathcal{D})$ becomes $p({\pmb{\theta}^*}|\mathcal{D})$.  (Sec. 3.3)
>
> Second, to model $p({\pmb{\theta}^*}|\mathcal{D})$, we use non-Bayesian pre-optimization for local regions of $\pmb{\theta}^*$ and propose a BL2O framework to estimate the local $p({\pmb{\theta}^*}|\mathcal{D})$.   (Sec. 3.4-3.7)
>
> **Related work**
>
> We clarify that Ortega et al. 2012 provided a way to directly model the uncertainty over the optimum through a Boltzmann-shaped posterior.  The posterior is iteratively updated with observed data through Kriging in Cao and Shen 2020.  Both ideas are adopted in our study.
>
> **Scattered argument**
>
> The original presentation of the statement was indeed very confusing.  We have now revised it as follows:
>
> The original Eq (3), now Eq (2) after revision, defines the problem of uncertainty quantification (UQ) for the quality of optimization solutions, which sets the goal for the rest of the paper.   To calculate the probability defined in Eq (2) and perform UQ, a direct albeit challenging way is to model the posterior over $\pmb{w}^*$ ($p(\pmb{w}^*|\mathcal{D})$) and then sample from the posterior.  When the optimizer $g$ is regarded fixed in existing literature, the posterior is actually  $p(\pmb{w}^*|\mathcal{D}, g)$.  A central contribution of ours is to further consider the optimizer as a source of uncertainty, model it as a random vector in an optimizer space, and perform posterior estimation of $p(\pmb{w}^*|\mathcal{D})$.
>
> **Ablation study**
>
> We appreciate the suggestion and performed ablation studies as suggested.  The new results are now in Appendix F.
>
> **Limitation of local neighborhood**
>
> We respectfully disagree that our model is more limited compared to parameterizing the optimizer space by hyperparameters of a specific algorithm.  The regions of interest are centered around parameters pre-optimized by non-Bayesian optimizers.  How local they are depends on the size of the region, just like the size of the grids for hyperparameters does.
>
> More importantly, parameterizing the optimizer space by hyperparameters of a specific algorithm has the following major drawbacks.  The resulting algorithmic space $\mathcal{G}$  is very limited and heavily depends on the specific optimizer. The $\mathcal{G}$ (a 1D space) parameterized by the hyperparameters of gradient descent is different from that (a 3D space) parameterized by the hyperparameters of Adam. In fact, each is a rather restricted region of the actual $\mathcal{G}$.  The intrinsic flexibility (uncertainty) that lies in an iterative optimizer's update rule is not explored at all in this parameterization. These drawbacks are empirically demonstrated in our experiments and fundamentally overcome by parameterizing the optimizer space using  parameters of neural networks.  More details can be found in the revised Sec. 3.3.
>
> **Quality of writing**
>
> Thank you for the suggestion. We hope that the current revision has been found to improve the quality of writing and the clarity of the methods.  We will continue to improve the writing quality thoroughly.

---

### Decision · Program_Chairs · 2021-01-07
**Final Decision**

**Decision:**

Reject

**Comment:**

The reviewers felt that the idea of learning a posterior distribution on optimization algorithms is very novel. However, the negative flip side of this novelty was that it was not clear how the prior and likelihood were defined so that Bayes rule could be approximated. The three reviewers appeared to find the paper somewhat confusing, and while the authors' made significant changes, it would be better to resubmit for a new set of reviews of the revised paper.